# Towards Full Delegation: Designing Ideal Agentic Behaviors for Travel Planning

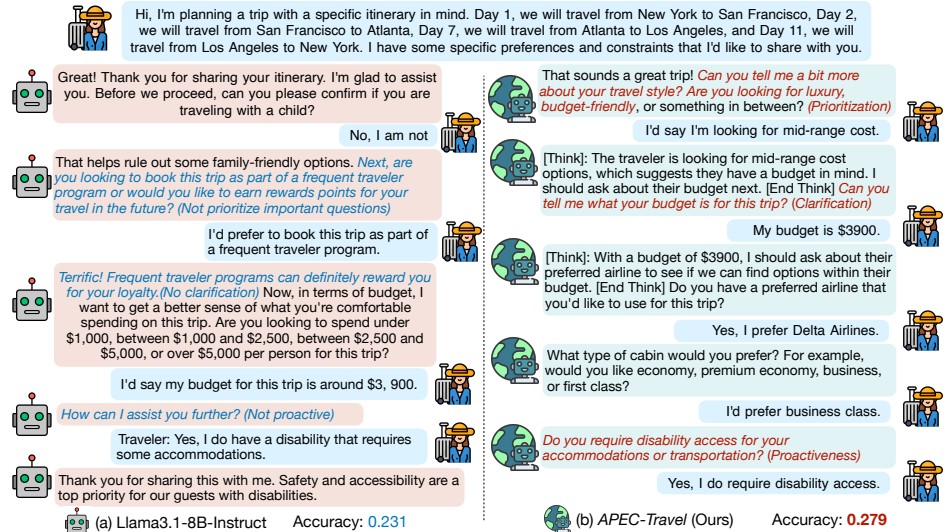

Figure 1: We develop `APEC-Travel`, a travel planning agent that effectively extracts hidden personalized preferences through multi-round dialogs with travelers. Compared to baseline models (left subfigure, (worse) behaviors *highlighted in blue*), `APEC-Travel` (right subfigure) *prioritizes* critical travel entries, asks for *clarification*, and *proactively* moves forward with new topics to gain more information (*highlighted in red*). These positive agent behaviors lead to improved accuracy in understanding personalized travel preferences.

## ABSTRACT

*How are LLM-based agents used in the future?* While many of the existing work on agents has focused on improving the performance of a specific family of objective and challenging tasks, in this work, we take a different perspective by thinking about *full delegation*: agents take over humans' routine decision-making processes and are trusted by humans to find solutions that fit people's personalized needs and are adaptive to ever-changing context. In order to achieve such a goal, the behavior of the agents, i.e., *agentic behaviors*, should be evaluated not only on their achievements (i.e., outcome evaluation), but also how they achieved that (i.e., procedure evaluation). For this, we propose APEC *Agent Constitution*, a list of criteria that an agent should follow for good agentic behaviors, including *Accuracy*, *Proactivity*, *Efficiency* and *Credibility*. To verify whether APEC aligns with human preferences, we develop `APEC-Travel`, a travel planning agent that proactively extracts hidden personalized needs via multi-round dialog with travelers. `APEC-Travel` is constructed purely from synthetic data generated by Llama3.1-405B-Instruct with a diverse set of travelers' persona to simulate rich distribution of dialogs. Iteratively fine-tuned to follow APEC Agent Constitution, `APEC-Travel` surpasses baselines by 20.7% on rule based metrics and 9.1% on LLM-as-a-Judge scores across the constitution axes.

## 1 INTRODUCTION

State-of-the-art Large Language Models (LLMs), such as GPT-4 (Achiam et al., 2023), Claude (Anthropic, 2024) and Llama (Touvron et al., 2023; Dubey et al., 2024) have been rapidly adopted by

users as chatbots, coding assistants or in place of traditional internet search. Current models are getting increasingly proficient at instruction-following with common post-training practices such as RLHF, RLAIF, etc. (Wei et al., 2021; Ouyang et al., 2022; Bai et al., 2022a).

However, there is still a non-trivial gap between instruction-following LLM and *agentic* LLMs. An LLM *agent* is a system which can execute *tasks* and take actions, such as using tools (Schick et al., 2024; Qin et al., 2023; Ocker et al., 2024), calling external APIs (Qin et al., 2023), writing code (Yang et al., 2024a), planning complex travel itineraries (Xie et al., 2024b), collaborating with other agents (Wu et al., 2023; Chen et al., 2023) or humans for general problem-solving. As intrinsic capabilities of pretrained LLMs have been improving with scaling, it becomes imperative to ask a meta-question: *What are the desired behaviors of LLM agents?*

We argue that enabling humans to ***delegate*** is a key property of LLM agents. Humans complete many tasks daily, weekly, monthly, or yearly that are repetitive or tedious such as ordering groceries, interacting with service providers or customer service, or planning vacations and trips. An autonomous LLM agent would be able to complete most or all of each of these tasks, only requiring human involvement for final approval if necessary.

To make progress towards this vision, we argue that LLM agents should be evaluated and optimized not only based on final outcome, e.g. success rate as is measured by current benchmarks, but also based on the procedure of how agents achieve the goal. In this paper, we take a broader view to evaluate such *agentic behaviors* and propose a set of principles, which we call the ***APEC Agent Constitution***:

- *Accuracy*. The quality of the final solution that the agent provides (e.g., number of questions that are answered correctly).
- *Proactivity*. Whether the agent proactively collects useful information to solve the task. Such information may be public or private, vague or precise, explicitly provided or inferred from requests.
- *Efficiency*. Whether the agent can achieve its goal with a minimal number of interactions (e.g., number of questions asked, API calls and tool uses).
- *Credibility*. The reliability with which agents achieve positive outcomes (e.g., amount of hallucination and inconsistency).

For each of these 4 axes, we develop quantitative measures so that it can be evaluated and/or optimized via various techniques such as RLAIF (Bai et al., 2022b) or RLHF (Ouyang et al., 2022). Compared to existing practices of evaluating LLM agents, which only focuses on the accuracy metric, our proposed `APEC` Agent Constitution allows evaluating the procedure by which an agent achieves outcomes. Furthermore, future work may expand `APEC` to cover an agent's ability to adapt to novel tasks or collaborate with other agents (Wu et al., 2023; Zhou et al., 2024).

As an instantiation of `APEC`, we investigate key research questions in the concrete agent task of Travel Planning (Xie et al., 2024b). We propose `APEC-Travel`, an agent optimized to proactively gather personalized travel preferences from a traveler through multi-round dialog. To create an agent that can be delegated with diverse travel requests, we create multi-round dialogs between travel agents and travellers with diverse backgrounds and implicit personalized preferences during travel planning. Using this synthetic dialog data, we fine-tune a traveller model as the environment for `APEC-Travel`. Then we sample outputs from `APEC-Travel` and iteratively improve `APEC-Travel` in terms of `APEC` using Direct Preference Optimization (Rafailov et al., 2023).

Through thorough experiments, we show that `APEC-Travel` achieves strong performance along our Agent Constitution `APEC`, and can infer hidden personalized travel requests with high accuracy. Figure. 1 illustrates an example of the agentic behaviors in which `APEC-Travel` excels.

## 2 RELATED WORK

**LLM-Powered Autonomous Agents.** LLMs have demonstrated remarkable reasoning and planning capabilities, leading to their wide adoption as the "brain" of agents across various domains (Wu et al., 2023; Li et al., 2023a; XAgent, 2023; Zhou et al., 2023b). LLM-powered agents significantly expand the boundaries of complex applications, including web interaction (Yao et al., 2022; Zhou

et al., 2023a; Deng et al., 2024; Zheng et al., 2024a; Koh et al., 2024), coding (Yang et al., 2024a; Jimenez et al., 2024; Trivedi et al., 2024; Yang et al., 2024b), embodied agents (Fan et al., 2022; Wang et al., 2023; Song et al., 2023), and social reasoning (Zhou et al., 2024; Kosinski, 2023; Shi et al., 2023; Xie et al., 2024a). These studies have primarily focused on integrating various external aids, such as specialized tools (Schick et al., 2024; Qin et al., 2023; Ocker et al., 2024) and symbolic solvers (Pan et al., 2023; He-Yueya et al., 2023; Liu et al., 2023), in order to enhance performance on complex tasks. However, we argue that an ideal autonomous agent should actively engage in multi-round dialogs to gather essential information from humans, tailoring its solutions to personalized contexts. In alignment with this perspective, the most closely related work to ours is Yao et al. (2024), where the agent interacts with human users to confirm decisions. Our work adopts a broader view, defining universal principles of agentic behavior. These human-like interactive agentic behaviors is crucial for building agents that can be fully trusted with delegated tasks.

**Alignment Fine-tuning.** Aligning LLMs with human preferences traditionally relies on human-annotated data for either building reward models (Ouyang et al., 2022; Bai et al., 2022a), or directly optimizing without explicit rewards (Rafailov et al., 2023; Meng et al., 2024; Amini et al., 2024). Recent approaches employ LLMs to annotate preferences via LLM-as-a-Judge prompting. Such AI-annotated preferences have shown performance on par with human labels (Bai et al., 2022b; Lee et al., 2024; Sun et al., 2024). The ready accessibility of AI-annotated preferences facilitates an iterative tuning paradigm, in which LLMs self-improve by labeling and learning from their own outputs. This process significantly reduces the costs associated with human annotation (Li et al., 2024; Yuan et al., 2024; Pang et al., 2024). Our work extends these efforts by combining objective metrics with LLM-as-a-Judge evaluations for preference labeling, achieving high task performance while still aligning with the general agentic behaviors.

**Travel Planning with LLMs.** Planning itineraries that satisfy all traveler constraints has proven challenging for even frontier LLMs (Xie et al., 2024b; Zheng et al., 2024b). Current efforts to enhance LLM performance on this task include fine-tuning (Bohnet et al., 2024) and hybrid approaches that integrate external tools (Xie & Zou, 2024) or solvers (de la Rosa et al., 2024; Hao et al., 2024) into the planning process. These methods often assume all constraints are explicitly provided, which is unrealistic as constraints typically emerge through multi-round dialogs between agents and travelers. A recent study (Zhang et al., 2024) bridges this gap by teaching agents to ask clarifying questions. However, human-like agentic behaviors entails more than simply seeking clarification. Our approach takes a step further by developing travel agents that adhere to the comprehensive Agent Constitution, enabling them to act as fully autonomous, human-like travel agents.

## 3 METHODOLOGY

Travel planning inherently involves multi-round task-oriented dialogs (TOD) and indirect requests (Mannekote et al., 2024). Unlike previous work in TOD that usually involves fine-tuning LLMs on human conversations with annotations (Hosseini-Asl et al., 2022) or rule-based data generation or augmentation (Samarinas et al., 2024; Li et al., 2023b), the proposed `APEC-Travel` is built purely from synthetic data following `APEC`. We first prompt a strong LLM (Llama3.1-405B-Instruct) to generate synthetic seed dialogs, which are used to initially supervised fine-tune `APEC-Travel` into a travel expert. Next, we iteratively train `APEC-Travel` based on preference-based optimization (DPO). In each iteration, `APEC-Travel` generates new dialogs, which are annotated with rewards given by rule-based objectives and LLM-as-a-Judge scores. `APEC-Travel` is then trained using these reward-annotated dialogs for the next-iteration preference optimization. This approach is scalable and addresses the challenge of data scarcity for building personalized LLM agents, reducing required human annotations of agentic behaviors. Figure. 2 shows an overview paradigm and workflow of `APEC-Travel`.

### 3.1 CONCRETE EVALUATION METRICS FOR `APEC`

The 4 axes in `APEC`, in particular *Proactivity* and *Credibility*, are defined at an abstract level. We perform several steps to derive concrete and quantitative metrics for them. First, we propose five candidate *neural* metrics that can be implemented via RLAIF: planning, prioritization, proactiveness, clarification, and helpfulness. We conduct human study to calibrate them with human judgement to

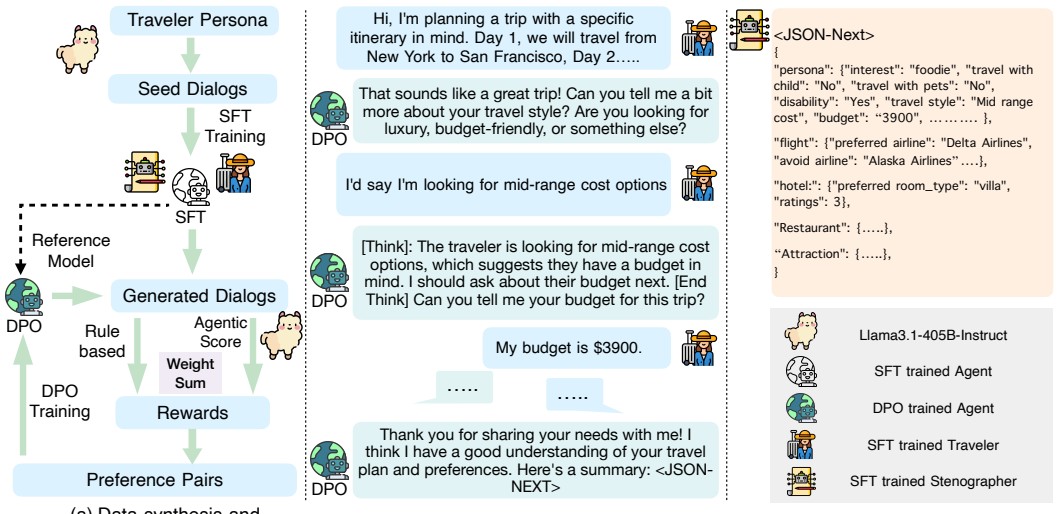

Figure 2: An overview of `APEC-Travel`. (a) We prompt Llama3.1-405B-Instruct to synthesize seed dialogs between a travel agent and travellers based on a diverse set of simulated traveller personas. These dialogs are used to fine-tune (SFT) Llama3.1-8B-Instruct, resulting in `APEC-Travel`-SFT. Next, `APEC-Travel` is trained with iterative Direct Preference Optimization (DPO), in which the latest `APEC-Travel`-DPO agent generates new dialogs with the traveller model in each iteration. These dialogs are ranked by a weighted combination of rule-based objectives and APEC scores assigned by a judge model (also Llama3.1-405B-Instruct). Note that the reference model is fixed as `APEC-Travel`-SFT throughout this process. (b) Overall workflow: `APEC-Travel` extracts traveler's personalized preference via multi-round dialog, after then the stenographer model summarizes the dialog into a symbolic representation (JSON).

make sure there are no substantial correlations among these candidate metrics, in order to make the measured axes of Agent Constitution independent of each other.

Then we further refine those candidate metrics with a small-scale human annotation. We select 200 examples from the seed dialogs and have human annotators evaluate them across these five metrics. Each metric is accompanied by a 5-point scale with detailed rubrics. To reduce variance, each dialog is reviewed by three annotators. 58 annotators from United States participate in our annotation process, with most annotators scoring between 10 to 15 dialogs. To ensure consistency and minimize variance, each dialog was annotated by 3 different annotators. Additionally, to further ensure reliability, annotators were asked to provide their rationale for each score assignment. More details about the human annotation process can be found in Appendix. A.4.

The correlation matrix between the five metrics is presented in Figure. 3. We make two observations: (1) a high correlation between the planning and prioritization scores and (2) helpfulness exhibits a non-trivial correlation with all other metrics, likely due to its subjective and non-concrete nature. Therefore, we choose to remove helpfulness and combined planning and prioritization together.

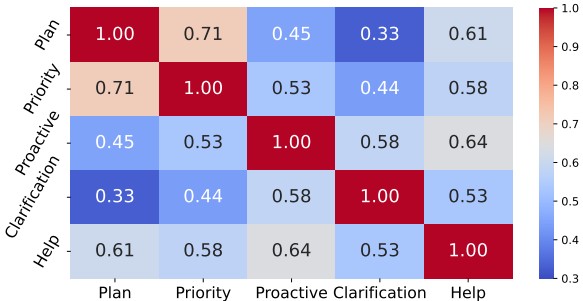

Figure 3: Correlation matrix from human annotation study examining the initial five agentic scores: Planning, Prioritization, Proactive, Clarification, and Helpfulness from which we derive the axes in `APEC`.

Finally, we propose concrete measure of each axis in Agent Constitution for travel planning as follows (also summarized in Table. 1):

*Accuracy.* Directly measures how much the agent has correctly understood the traveler's requests from diverse background. While there are a lot of dimensions for travelers' preferences, each traveler has critical entries that are way more important than others and thus we use a weighed distance.

*Efficiency*. We constrain the dialog to be short (in terms of number of rounds) and check whether the agent has successfully obtained critical information given the limited dialog exchange, measured by the symbolic metric. We expect the agent to quickly navigate towards most important information, by inferring the hidden persona based on all information given by the traveler (e.g., their initial request, talking styles, etc). In addition, we also leverage neural metric "plan & priority" to evaluate the procedure efficiency achieved by the agent, which is more subjective.

| Metric | Agent Constitution | | | |
|---|---|---|---|---|
| | Accuracy | Proactiveness | Efficiency | Credibility |
| Symbolic (JSON) | ✓ | | ✓ | |
| Plan & Priority | | | ✓ | ✓ |
| Proactive | | ✓ | | |
| Clarification | | | ✓ | |

Table 1: Evaluation protocol for agentic behavior in travel planning. Plan & Priority, Proactive and Clarification are assigned by LLMs (Llama3.1-405B-Instruct in this work).

*Proactivity* and *Credibility*. We measure the two axes using corresponding neural metrics "proactiveness" and "clarification". The criterion is whether agents actively propose critical questions, clarify any ambiguity, some of which the traveler may not even be aware of, and demonstrate a streamlined thinking process.

**Reward Construction.** With the above metrics, we build a reward objective $R$ that strikes a balance between achieving accurate task-specific objectives and adherence to agentic behaviors. This reward is used to evaluate a dialog driven by `APEC-Travel`. Specifically, we have:

*Accuracy $R_c$*. We check the accuracy of the travel preferences inferred by `APEC-Travel` at the end of the dialog against the ground truth preference specified in the persona.

*Agentic Score $R_a$*. We use the LLM-as-a-Judge approach ([Bai et al., 2022b](#)) to assign scores on plan & priority, proactive and clarification of a given dialog, and then calculate the sum of these scores as the overall agentic score $R_a$ for each dialog. The prompts for obtaining agentic behaviors scores are detailed in Appendix. [A.5](#).

The final reward score $R$ is a linear combination of accuracy and agentic score, with a hyperparameter $\alpha \in (0, 1)$ to balance them, simply formalized as:

$$R = \alpha R_c + (1 - \alpha)R_a. \tag{1}$$

## 3.2 TRAVELER SIMULATION

**Persona Simulation.** To simulate diverse traveler personalities, we represent each traveler from two categories of information: *persona* (characteristics related to travel, such as travel-oriented interests) and *travel constraints* (traveler's personalized requirements or preferences, such as preferred airline and need for disability access, etc.). We create a set of 54 entries that capture these two perspectives and randomly assign values to each entry to generate a variety of synthetic travelers. We create three disjoint traveler sets for seed dialog generation (10k examples), reward construction (10k examples), and final evaluation (1k examples) respectively. Details of the 54 persona entries can be found in Appendix. [A.1](#).

**Critical Persona Entry Selection.** In reality, each traveler has a unique set of priorities related to their travel preferences. For instance, a traveler with a disability may prioritize accessible services above all else. Accordingly, travel agent's success can be assessed by whether or not an agent identifies the most critical persona entries for each individual traveler.

We ensure realistic critical persona entries by factoring in basic user characteristics (such as age, job, education, marital status, disability, and travel style) along with empirical traveler categories (the product of an online survey of 1385 travelers). We create the critical entries for a given user by prompting Llama3.1-8B-Instruct to rank the importance of the full set of persona entries and select the top 20% as critical entries. Our agent will seek to identify these through multi-round dialog. The full prompt and details of the survey can be found in Appendix. [A.2](#).

## 3.3 SEED DIALOG GENERATION

With each unique traveler persona and their critical entries, we then synthesize multi-round dialogs between travel agents and the traveler as seed data. Each dialog consists of multiple turns of conversation in which a travel agent predicts the traveler's most critical entries and plans a series of questions to collect information from the traveler (see Figure. 2 (b) for an example).

To make the multi-round dialogs more realistic, we consider a three-role setting: Agent, Traveler and the Stenographer, They share the collaborative goal to reproduce the traveler's preference: 1) The *agent's goal* is to fully commitment to the Agent Constitution and proactively seek information that completes the travel requests. In addition, agent should logically "think" what should be a good next question based on the collected information from the traveler. This reasoning process is tagged with [Think] tokens. 2) The *traveler's goal* is to be faithful to their persona and represent them clearly in the dialog. We randomly assign the traveler's chat style (e.g., casual, wordy, etc.) to increase the diversity of each dialog. 3) The *stenographer's goal* is to translate the dialog into a symbolic (JSON) representation of the traveler's persona. These principles are incorporated into our prompt for Llama3.1-405B-Instruct to synthesize the seed dialogs, with full details in Appendix A.3.

## 3.4 TRAINING

**SFT.** In the SFT stage, we use the seed dialogs to train three distinct roles: agent (i.e., APEC-Travel), traveler and stenographer, as outlined in Figure. 2. The agent model is trained to predict the next question along with its own reasoning stage, denoted by [Think] and [Think_end] tags within the same utterance; the traveler model is trained to answer the agent's questions based on the traveler persona; while the stenographer is to summarize the dialogs between agent and traveler, and reconstruct a symbolic representation (JSON) of the travel preferences.

**Iterative DPO.** To obtain preference data for Direct Preference Optimization (DPO) (Rafailov et al., 2023) training of the agent model, we use the reward function described in Section 3.1. Specifically, for a given prompt, we generate two different dialogs using the SFT-tuned agent and traveler model, as well as their associated JSON outputs from the stenographer model. We then assign a reward to each dialog according to Equation. 1 and use the relative scores to determine the preferred and rejected responses. During DPO training, we mask the loss on tokens from the traveler model. Additionally, for each iteration of DPO, we use the same SFT-tuned model as reference model. We compared this training approach with the alternative approach of using the last DPO iteration's model as reference model (Pang et al., 2024; Yuan et al., 2024) in Section. 4.3.

## 4 EXPERIMENTS

### 4.1 EXPERIMENT SETUP

**Training.** We have two training stages: SFT and iterative DPO, as described in Section. 3.4. The three models (agent, traveler and stenographer) are initialized from Llama3.1-8B-Instruct. When generating dialogs for iterative DPO training, we sample from these three models via vLLM (Kwon et al., 2023). We set temperature=1.0 for the agent model to boost diversity in agent conversations. The full configs of SFT and DPO training can be found in Appendix. A.7.1 and Appendix. A.7.2.

**Evaluation.** We evaluate how APEC-Travel adheres to Agent Constitution on a held-out set of 1k examples without overlapping persona with the seed or training data. We consider three aspects: accuracy, efficiency and agentic scores following Agent Constitution (Sec. 3.1). The scoring prompt can be found in Appendix. A.5.

**Baselines.** We evaluate APEC-Travel by comparing with Llama3.1-8B-Instruct, and a much stronger LLM, Llama3.1-70B-Instruct. We consider both plain and reasoning (with self-thinking ) prompting strategies. The prompt for baselines can be found in Appendix. A.6.

| Model | Average | Accuracy | | Efficiency | | Agentic Scores (Full score in each axis: 5) | | | |
|---|---|---|---|---|---|---|---|---|---|
| | #Rounds | Overall | Critical | Overall | Critical | Plan & Priority | Proactive | Clarification | Total |
| Llama-3.1-8B | 15.49 | 0.231 | 0.301 | 0.015 | 0.019 | 3.88 | 4.07 | 3.90 | 11.86 |
| Llama-3.1-8B-Reasoning | 15.50 | 0.217 | 0.287 | 0.014 | 0.018 | 3.80 | 4.06 | 3.87 | 11.75 |
| APEC-Travel-SFT | 9.39 | 0.261 | 0.417 | 0.029 | **0.047** | 4.46 | 4.25 | 3.68 | 12.41 |
| APEC-Travel-DPO | | | | | | | | | |
| Iteration 1 | 11.19 | 0.286 | 0.423 | 0.027 | 0.041 | 4.36 | 4.22 | 3.86 | 12.46 |
| Iteration 2 | 9.77 | 0.279 | 0.425 | **0.031** | **0.047** | **4.48** | **4.32** | **4.13** | **12.95** |
| Iteration 3 | 11.18 | 0.295 | 0.442 | 0.029 | 0.044 | 4.35 | 4.30 | 3.99 | 12.67 |
| Iteration 2+3 | 11.36 | **0.296** | **0.448** | 0.028 | 0.043 | 4.44 | 4.28 | 3.79 | 12.52 |
| *Other SoTA LLMs* | | | | | | | | | |
| Llama3.1-70B | 15.49 | 0.243 | 0.308 | 0.016 | 0.020 | 3.95 | 4.19 | 3.84 | 12.00 |
| Llama3.1-70B-Reasoning | 15.48 | 0.229 | 0.310 | 0.015 | 0.020 | 3.93 | 4.31 | 4.02 | 12.28 |

Table 2: Performance of APEC-Travel compared with baselines on test set. The term "Iteration 2+3" means we mix the training dialogs of iteration 2 and iteration 3. Both baseline models, Llama3.1-8B and Llama3.1-70B, are instruction-tuned models; and we set the reward controller $\alpha$=0.1 throughout this experiment.

## 4.2 MAIN RESULTS

**Overall Results.** We compare APEC-Travel with baseline models and report the results in Table. 2. Our results demonstrate significant and consistent improvements of APEC-Travel over both the baseline models (Llama-3.1-8B-Instruct) and larger LLMs (Llama-3.1-70B-Instruct) across various metrics in all three axes. Specifically, iterative DPO training after SFT reaches optimal performance after two iterations, in which both efficiency and agentic scores are the highest. It is important to emphasize that efficiency, defined as accuracy gain per round, accurately reflects an agent's intelligence in inferring traveler preferences. Therefore, the 2-iteration DPO-trained APEC-Travel engages in more concise dialogs (averaging only 9.77 rounds) but achieves a final accuracy comparable to other models requiring much longer dialogs.

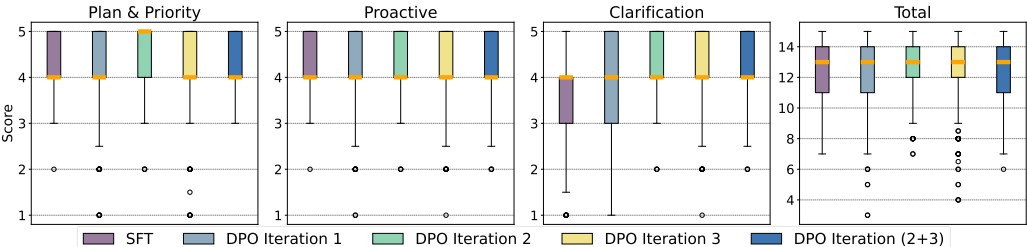

Figure 4: Breakdown of agentic scores across all axes. We compare APEC-Travel-SFT and APEC-Travel-DPO for each axis. The *median* of each box plot is highlighted in orange. Axes from left to right: Plan & Priority; Proactive, Clarification and the Total of these three axes.

**Individual Agentic Behaviors.** In addition to the agentic scores in Table. 2, we also detail the score distribution for each agentic axis in Figure. 4. The results demonstrate that the iteratively trained APEC-Travel-DPO, especially in iteration 2, significantly enhances the agentic scores across the entire test set. This improvement underscores the effectiveness of our training recipe in consistently aligning agents with the principles in the Agent Constitution.

## 4.3 ABLATION STUDIES AND ANALYSES

**Synthetic Seed Data Quality.** Constructing high-quality seed dialog data is crucial for building travel agents that adhere to the Agent Constitution. To understand the role of Llama3.1-405B-Instruct-bf16 model in synthetic data quality, we compare its agentic scores with the quantized Llama3.1-405B-Instruct-FP8. As the results in Figure. 5 show, the 405B-Instruct-bf16 model significantly outperforms the FP8 counterpart in generating high-quality synthetic data. Specifically, the seed data generated by

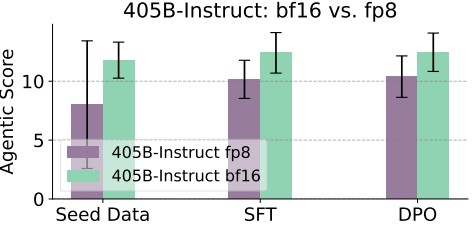

Figure 5: Comparison of the *Total* agentic scores across all axes for Llama3.1-405B-Instruct BF16 versus FP8. From left: seed data, APEC-Travel-SFT, and APEC-Travel-DPO.

bf16 scores 3.78 points higher (out of a full score of 15) than FP8. Consequently, agent models trained on bf16-generated data scored 2.25 (SFT) and 2.07 (DPO) points higher. Although quantization models are known to achieve comparable results to the original model in some reasoning benchmarks (Peng et al., 2023), our empirical ablation study demonstrates that a strong model is essential for synthesizing high-quality data.

**How Reward Construction Influences DPO.**
DPO training is highly sensitive to how the reward is constructed. To investigate the reward's effects on DPO training, we conduct an ablation study by varying the controller hyperparameter $\alpha$, which balances the accuracy $R_c$ and the agentic score $R_a$. Specifically, we set $\alpha \in [0, 0.05, 0.1, 0.15, 0.2, 0.25, 0.3]$ and present the corresponding metrics on the test set in Figure. 6. Our results indicate that increasing the emphasis on accuracy (i.e., a larger $\alpha$) generally enhances both the overall and critical accuracy metrics, while the agentic scores tend to decrease gradually. This pattern suggests that the final performance of the DPO training is overall aligned with the components in the reward objective. Note that we keep the $\alpha$ consistent in across all the DPO iterations.

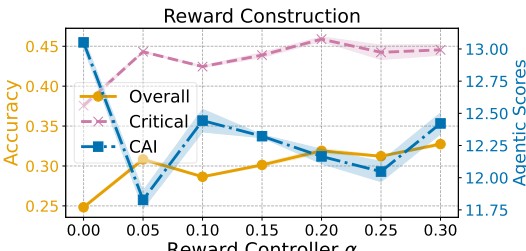

Figure 6: Performance across varied weight combinations, controlled by the hyperparameter $\alpha$, in the reward construction for DPO training. We report accuracy, including overall accuracy in inferring traveler preferences and accuracy on critical entries, as well as the *Total* agentic score.

**Effectiveness of Agent's Reasoning.** We evaluate the impact of the reasoning process (i.e., the agents' self-thinking process highlighted by the `[Think]` token) in `APEC-Travel`. Specifically, we remove the instructions for the `[Think]` reasoning from the original seed data synthesis prompts (refer to Appendix. A.3) to generate contrastive dialogs without the `[Think]` process. We then compare an SFT-trained agent using these modified dialogs to the original SFT agent trained with dialogs that include the `[Think]` process. As shown in Table. 3, the self-thinking reasoning process clearly improves all agentic scores, underscoring its essential role in improving agentic behaviors for `APEC-Travel`.

|  | w/o → w/ `[Think]` |
|---|---|
| Plan & Priority | 4.18 → **4.46** |
| Proactive | 4.01 → **4.25** |
| Clarification | 3.58 → **3.68** |
| Total | 11.79 → **12.41** |

Table 3: Comparison between `APEC-Travel`-SFT trained with and without intermediate reasoning process (tagged by `[Think]`).

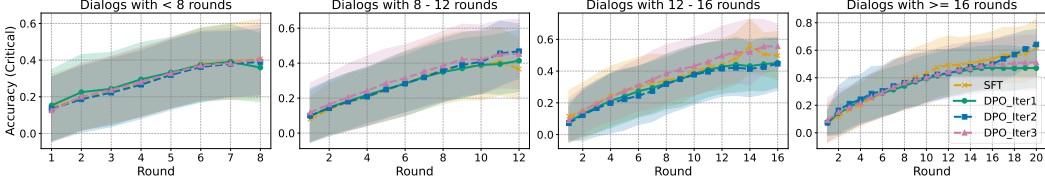

Figure 7: Accumulated accuracy of critical entries across dialog rounds. Dialogs are categorized into 4 groups based on number of rounds: <8, 8 to 12, 12 to 16, and >16. We report average (dashed line) and standard deviation (shaded area) on test set for both SFT and DPO trained models.

**Efficiency in Inferring Critical Entries.** Table. 2 presents the overall efficiency of `APEC-Travel` throughout the dialog rounds. To gain a deeper understanding of how fast `APEC-Travel` identifies critical traveler preferences, we break down the accuracy gain across dialog rounds and report the accumulated accuracy in Figure. 7. Our results demonstrate that DPO trained models exhibits a faster gain in critical accuracy during the early rounds compared to those trained with SFT. This highlights the essential role of DPO in our training recipe - enhancing `APEC-Travel`'s capability of prioritizing more critical entries for each traveler.

|  | Fixed | Recursive |
|---|---|---|
| Accuracy |  |  |
|   - Overall | 0.279 | 0.320 |
|   - Critical | 0.425 | 0.458 |
| Efficiency |  |  |
|   - Overall | 0.031 | 0.023 |
|   - Critical | 0.047 | 0.033 |
| Agentic (Total) | 12.95 | 12.10 |

Table 4: Comparison between two different iterative DPO training paradigms: "Fixed" uses SFT as the reference model with dialogs generated by the new model in each iteration; "Recursive" employs dialogs from each iteration's model to train the same model cyclically.

| Error Type | Number of Examples |
|---|---|
| Agent model |  |
|   - Fail to ask meaningful questions | 1 |
|   - Limited dialog rounds | 3 |
| Traveler model |  |
|   - Hallucination | 3 |
|   - Answer wrong question | 2 |
| Stenographer model |  |
|   - Hallucination | 5 |
|   - Wrong format | 3 |
| Low-quality simulated persona | 3 |

Table 5: Statistics of error types. These 20 examples are randomly selected from those with low accuracy scores (overall < 0.15 & critical < 0.3). The agent model is DPO-trained model (Iteration 2).

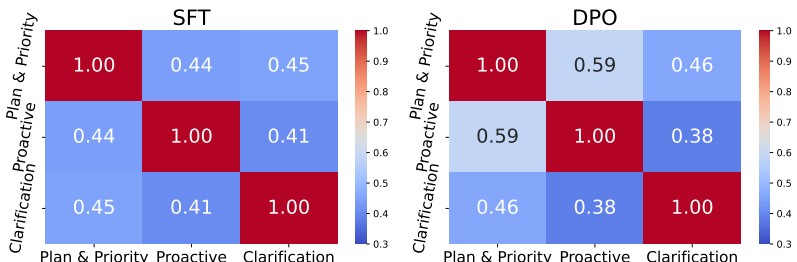

Figure 8: Correlation matrix of the three agentic scores: Plan & Priority, Proactive and Clarification. Left: `APEC-Travel`-SFT; Right: `APEC-Travel`-DPO (Iteration 2).

**Iterative Training Paradigm.** Iterative DPO training is crucial for aligning our agent with the Agent Constitution. To explore potential improvements, we compare our "Fixed" iterative DPO paradigm (where the reference model is fixed with the SFT model, while training data dialogs are sampled from the last round of DPO model) with the "Recursive" iterative training paradigm (where the DPO model from the previous iteration is used as the reference model) (Yuan et al., 2024; Wang et al., 2024). As is shown in Table. 4, the "Recursive" paradigm improves final accuracy (both overall and critical) but significantly reduces efficiency and agentic scores. Additionally, we observe that the "Recursive" training increases the average number of dialog rounds to 15.44, which is considerably higher than the 9.77 rounds observed in the "Fixed" paradigm. We speculate that the "Recursive" paradigm mostly optimizes towards the final accuracy.

**Orthogonality of Agentic Scores.** To examine the relationship between the three agentic scores—Plan & Priority, Proactive, and Clarification—we analyze their correlations in dialogs from both the SFT and DPO (Iteration 2) models, as shown in Figure. 8. Our results indicate that these scores are generally not highly correlated, suggesting that they evaluate `APEC-Travel` from distinct perspectives in relation to becoming a fully-delegated agent.

### 4.4 ERROR ANALYSIS: INSTANCES OF LOW ACCURACY IN `APEC-TRAVEL`

To gain deeper insight into the circumstances under which our agent underperforms, we conduct error analysis on all three models involved: the agent, the LLM-simulated traveler, and the stenographer. Specifically, we randomly select 20 examples with low accuracy scores (overall < 0.15 & critical < 0.3) and summarize the reasons for these errors in Table. 5. We observe that a significant number of low accuracy examples (13 out of 20) are due to errors from either the traveler model or the stenographer model, especially the stenographer model sometimes failed to generate valid JSON. Specific to the agent model, a common error is generating short dialog, which restricts the total number of preference entries.

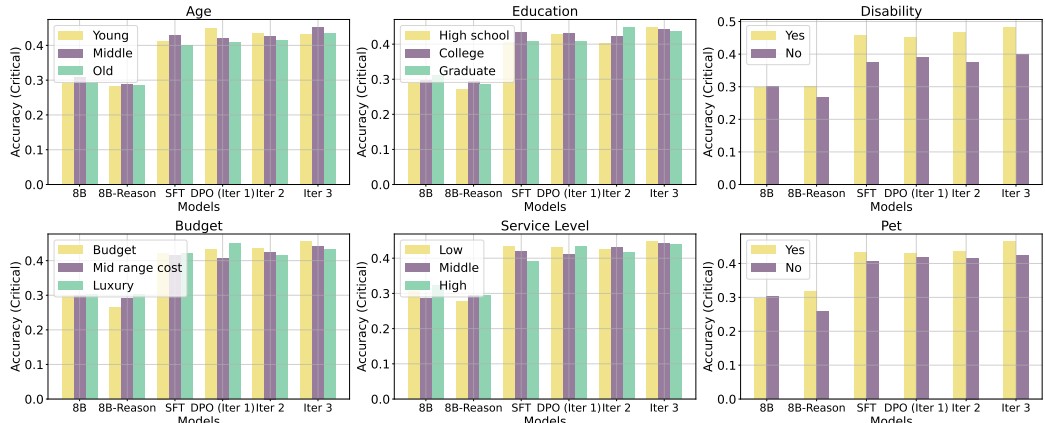

Figure 9: Performance (correctness of critical entries) breakdown based on six traveler personas: Age, Education, Disability, Budget, Service Quality, and Travel with Pet.

### 4.5 PERSONALIZATION: IS APEC-TRAVEL ROBUST ACROSS DIFFERENT PERSONAS?

Each traveler is simulated to have their unique persona and corresponding critical travel entries (Section. 3.2). To evaluate whether APEC-Travel effectively accommodates personalized personas, we analyze its performance based on user personas. Specifically, from the 54 travel persona entries, we select 6 important ones, including age, education, disability, budget, service quality, and traveling with pets. The accuracy of these critical entries is shown in Figure 9. Our results indicate that APEC-Travel is generally robust to various traveler types and consistently outperforms the baselines. Notably, APEC-Travel achieves better results in critical entry accuracy for travelers with disabilities than those without. This is because for travelers with disabilities, certain preferences, such as the need for accessible flights, become critical. This enhancement in performance demonstrates that APEC-Travel is adept at addressing the personalized needs of these travelers.

## 5 CONCLUSION

We envision the future of agents as *full delegation*, where humans regularly delegate their routine decision-making to agents who can comfortably make high-quality decisions in real-world scenarios tailored to personal needs. To move towards this goal, in this work, we propose APEC, Agent Constitution that describes principles of the desired agentic behaviors. We instantiates our vision in the specific task of Travel Planning, which requires agents to proactively collect each traveler's personal needs via multi-round dialogs, and develop a method to optimize towards APEC using synthetic data and iterative self-training. The resulting agent, APEC-Travel, achieves improvement over baseline both in terms of accuracy (+20.7%) and agentic scores (+9.1%). Recent work (Hao et al., 2024; Ju et al., 2024) demonstrates that symbolic solvers, such as MIPS solvers, can be utilized to generate travel itineraries that satisfy constraints described in JSON format. Therefore, converting our inferred traveler's persona JSON into a personalized itinerary using a dedicated symbolic solver would be a valuable next step, which we plan to explore in future work.

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

# A  APPENDIX

## A.1  DETAILS OF PERSONA SIMULATION

| Attraction | Choices |
|---|---|
| preferred type | museum, art, park, shopping, landmark, historical site, zoo, aquarium, no preference |
| preferred duration | Less than 1 hour, 1-2 hours, 2-3 hours, More than 3 hours, no preference |
| popularity | Low, Medium, High, no preference |
| ratings | 1-5 |
| need disability access | Yes, No |
| need guided tours | Yes, No, no preference |
| special events | Concerts, Festivals, Workshops, Lectures, no preference |
| preferred amenities | Food and drink, Restrooms, Gift shop, Wi-Fi, no preference |
| **Flight** | **Choices** |
| preferred airline | American Airlines, United Airlines, Delta Airlines, Alaska Airlines, JetBlue Airways, no preference |
| avoid airline | American Airlines, United Airlines, Delta Airlines, Alaska Airlines, JetBlue Airways, no preference |
| preferred cabin | economy, coach, business, no preference |
| preferred refundablity | refundable, non-refundable, no preference |
| preferred fly time | morning, afternoon, red-eye, no preference |
| preferred meal options | vegetarian, gluten-free meals, no preference |
| preferred change policies | free changes, change for a fee, no preference |
| need in-flight Entertainment | free Wi-Fi, paid Wi-Fi, no preference |
| preferred aircraft type | boeing, airbus, no preference |
| avoid aircraft type | boeing, airbus, no preference |
| need disability access | Yes, No |
| need travel insurance | Yes, No |
| **Hotel** | **Choices** |
| preferred room type | entire home, private room, suite, villa, no preference |
| preferred house rules | No parties, No smoking, No children under 10, No pets, Quiet hours, no preference |
| preferred brand | Hilton, Marriott, Hyatt, IHG, Accor, Best Western, Choice Hotels, no preference |
| avoid brand | Hilton, Marriott, Hyatt, IHG, Accor, Best Western, Choice Hotels, no preference |
| preferred proximity | downtown, airport, beach, city center, public transportation, no preference |
| ratings | 1-5 |
| preferred amenities | Wi-Fi, Breakfast, Fitness center, Pool, Parking, no preference |
| preferred services | Room service, Concierge, Laundry, Tour desk, no preference |
| preferred cancellation policy | Flexible, Moderate, Strict, no preference |
| preferred room features | Air conditioning, TV, Mini-bar, Safe, no preference |
| preferred bathroom features | Shower, Bathtub, Hair dryer, Toiletries, no preference |
| need disability access | Yes, No |
| **Persona** | **Choices** |
| job | student, software engineer, researcher, banker, teacher, artist, entrepreneur, retiree, doctor, lawyer, sales, marketing manager, journalist, small business owner, government employee |
| age | 18-70 |
| interest | museum, music event, sport games, hiking, foodie, beach, city tour, adventure sports |
| education | High school, College, Graduate |
| marital status | Single, Married, Divorced |
| travel with child | Yes, No |
| travel with pets | Yes, No |
| travel style | Budget, Mid range cost, Luxury |
| travel frequency | Rarely, Occasionally, Frequently |
| disability | Yes, No |
| budget | 500-10000 |
| **Restaurant** | **Choices** |
| preferred cuisines | Tea, Pizza, French, Bakery, Seafood, Italian, Chinese, Indian, Japanese, Korean, no preference |
| ratings | 1-5 |
| preferred dining style | Casual, Formal, Buffet, Food truck, no preference |
| preferred seating options | Indoor, Outdoor, Takeout, Delivery, no preference |
| preferred payment methods | Cash, Credit card, Mobile payment, no preference |
| can reservations | Yes, No |
| preferred parking options | Street parking, Parking lot, Valet parking, no preference |
| pet friendly | Yes, No, no preference |
| allow smoking | Yes, No, no preference |
| need live music | Yes, No |
| need disability access | Yes, No |

Table 6: The fields and their respective possible values for personal generation.

**Persona Entries.**

We list all the travel entries involved in the traveler simulation (Section. 3.2) in Table. 6.

## A.2  DETAILS OF CRITICAL PERSONA ENTRY SELECTION

**Empirical Traveler Survey.**  We conduct an online survey of 1385 travelers to give our synthetic personas an empirical foundation. Participants were screened from a broad US-based pool who responded that they travel four or more times per year. Survey participants were asked to evaluate the quality of model-derived travel using several hand-crafted factors. The results are summarized in Table. 7:

| Factor | Share (%) |
|---|---|
| Total price | 23.9 |
| Specific level of service (e.g., hotel stars, airfare class) | 17.7 |
| Simple or few steps | 15.8 |
| Value per dollar | 14.1 |
| Travel at preferred time | 11.1 |
| Minimum time in transit | 9.0 |
| Travel or stay with preferred brands | 8.4 |

Table 7: Factors and percentages of different types of travelers identified in human interviews. The factors are provided by travelers as primary concerns when assessing the quality of itineraries.

**Critical Entry Ranking Prompt.**

We use the following prompt to select personalized critical entries for each traveler via Llama3.1-8B-Instruct.

> **Critical Entry Ranking Prompts**
>
> You are travel expert and understand traveler's customized travel preference based on their personas very well. Given a traveler who is a {age}-year-old {job} with {education} degree, {marital_status} marital status, {disability}, and has {travel_style} travel style. This traveler is also very concerned about their {empirical}, please rank the following fields in order of importance for this traveler. The fields are related to their personal profile, flight requirements, hotel requirements, restaurant preferences, and attraction interests. The list is:
>
> {initial_list}
>
> Please return a JSON in the end, in which the key is rank, and value is the field. The final ranked JSON is:

## A.3  PROMPT OF SEED DIALOG GENERATION

> **Prompts for Training Dialog Synthesis**
>
> You're a world-level simulation writer. You are simulating a conversation between a travel agent and a traveler. The traveler has specific personality traits, travel constraints, and preferences, which are partially detailed in the JSON below (Persona and Preferences JSON). Most fields are marked as UNKNOWN in the beginning.
>
> Persona and Preferences JSON:
>
> <JSON-NEXT>
>
> {empty_json}
>
> Some of the fields are more critical than others, but the critical fields are customized to each traveler. The critical fields for this traveler are:
>
> Critical fields list: {critical_fields}.
>
> The value of each field is UNKNOWN to the travel agent in the beginning. The travel agent's most important goal is to figure out the critical fields following the order in critical fields list through

a structured and professional conversation. It is highly rewarded if the travel agent can also ask non-critical fields from the Persona and Preferences JSON based on all given information as well. But it's ok to leave some non-critical fields UNKNOWN in the final.

We also have a JSON that describes the traveler's ground-truth value of each travel field in the following. Note this is not directly accessible to the travel agent. The travel agent can only uncover fields in the JSON by asking questions to the traveler.

Traveler's ground-truth JSON:

<JSON-NEXT>

{ground_truth_json}

Make sure following the rules below during conversation simulation:

Conversation Structure:

1) Complete Simulation: The language model should simulate the entire conversation from start to finish without prompting the user for responses. This rule must be strictly adhered to.
2) Initial Greeting: The travel agent should start the conversation with a simple professional greeting, such as "Hello! How can I assist you with your travel plans today?" The greeting should not include any specific information about the traveler.
3) Traveler's Initial Request: The traveler should describe their personality and travel preferences, including any constraints. The description should not include personal details such as age, job, education, or marital status. If the traveler specifies an itinerary, it should be formatted as "Day X, we will travel from X to Y".
4) Targeted Questions: The travel agent should ask questions that each aim to clarify only one specific field based on the Traveler's ground-truth JSON. The traveler should respond only with information relevant to the question asked, without volunteering additional details.
5) Silent Self-Thinking [Think]: During the conversation, the travel agent may engage in silent self-thinking moments to internally process the information provided by the traveler. These moments should be labeled with a [Think] tag. The agent should not verbalize these thoughts but use them to guide the next question or comment. This internal reflection helps in making informed decisions about what information to seek next, ensuring the conversation remains focused and relevant.
6) Conversation Conclusion and JSON Summary: The conversation should end with the travel agent summarizing the updated travel plan and preferences. After the summary, the agent should list the complete Persona and Preferences JSON, marking fields as "UNKNOWN" if they were not discussed or clarified during the conversation. The JSON should be introduced with a <JSON-NEXT> token.
7) Format: No need to have any round indicators. If a message is from travel agent, start with "Travel Agent:", if it is from traveler, start with "Traveler:". If it is a [Think], start with "Travel Agent [Think]:"

Interaction Style:
1) The travel agent is professional, proactive, and helpful, aiming to provide personalized service.
2) The traveler is a {traveler_style} {job}. Ensure the conversation aligns with the traveler's meticulous nature.
3) The traveler is detailed and clear in their responses, facilitating a smooth information exchange.
Please simulate the entire conversation simulation. The dialog is better to have more than 10 rounds. That would be a great one. Please generate the entire conversation:

## A.4 DETAILS OF HUMAN ANNOTATION ON DIALOGS

To find out the an effective and efficient set of metrics to evaluate a dialog between `APEC-Travel` and the traveler, we prepare 200 dialogs and let human annotators to evaluate them. In this run, we have designed 5 metrics: planning, prioritization, proactiveness, clarification and helpfulness. We ask the annotators to grade each dialog according to the following rubrics:

---

**Rubrics of Dialogs Human Annotation**

Rate the generated dialog from the agent on a scale of 1 to 5, using the following scoring criteria:
- Agent behavior:
[Planning] Is the question plan generated by agent at the beginning reasonable and the order logically correct?
5: The agent's initial question plan is comprehensive, covering all necessary aspects logically and efficiently.
4: The question plan is mostly reasonable, covering most necessary aspects with minor logical gaps.
3: The plan addresses some necessary aspects but lacks a logical flow or misses key areas.
2: The plan is vague, addressing only a few necessary aspects without a clear logical order.
1: There is no clear plan or logical order in the questions asked.

[Prioritization] Does the travel agent follow the question plan and prioritize more important questions?
5: The agent strictly follows the question plan and effectively prioritizes questions based on their importance and relevance to the user's needs.
4: The agent generally follows the question plan and prioritizes important questions, with minor deviations.
3: The agent occasionally follows the question plan but often fails to prioritize important questions.
2: The agent rarely follows the question plan and frequently misprioritized questions.
1: The agent does not follow any discernible plan or prioritization.

[Proactive] Does the agent ask good proactive questions to understand the user's preference?
5: The agent consistently asks insightful proactive questions that reveal deep understanding of the user's preferences.
4: The agent asks proactive questions that are generally relevant but could be more insightful or targeted.
3: The agent occasionally asks proactive questions, but they often miss the mark or are too generic.
2: The agent rarely asks proactive questions, and when they do, they are not relevant or useful.
1: The agent does not ask any proactive questions to understand the user's preferences.

[Clarification]Does the agent ask clarification questions if the traveler's response is vague?
5: The agent always asks for clarifications when responses are vague, ensuring complete understanding.
4: The agent usually asks for clarifications on vague responses, but may miss some opportunities.
3: The agent sometimes asks for clarifications, but often proceeds without full clarity.
2: The agent rarely seeks clarifications, leading to misunderstandings or incomplete information.
1: The agent never asks for clarifications, regardless of the clarity of the user's responses.

[Helpfulness] Is the agent generally helpful towards the traveler?
5: The agent is extremely helpful, providing accurate, relevant, and complete information aligned with the user's needs.
4: The agent is mostly helpful, providing generally relevant information with minor inaccuracies or omissions.
3: The agent provides some helpful information, but there are significant gaps or inaccuracies.
2: The agent provides minimal helpful information, with major inaccuracies or irrelevance.
1: The agent provides no helpful information or guidance.

---

## A.5 PROMPT FOR LLM-AS-A-JUDGE SCORING

According the the human annotation in Appendix. A.4, we find that planning and prioritization are highly correlated while helpfulness shares correlation with all other metrics Figure. 3. Therefore, we merge planning and prioritization and remove the helpfulness as our final metrics. With them, we present the prompt used for the LLM-as-a-Judge scoring in the following, where we use Llama3.1-405B-Instruct to assign agentic scores to each dialog.

---

**Prompt for LLM-as-a-Judge Scoring**

Review the conversation between travel agent LLM and travelers with diverse personalities, rate how well the agent gradually figures out the traveler's customized characteristics and travel request and rate to what degrees the agent asked a question that demonstrated good agentic behavior.

Specifically, rate the dialog from the agent on a scale of 1 to 5, using the following scoring criteria:

- Agent behavior:

Planning and Prioritization: Does the travel agent ask questions in a logically correct order and prioritize more important questions?
- 5: The agent strictly follows a reasonable question plan and effectively prioritizes questions based on their importance and relevance to the user's needs.
- 4: The agent generally follows a reasonable question plan and prioritizes important questions, with minor deviations.
- 3: The agent occasionally follows a reasonable question plan but often fails to prioritize important questions.
- 2: The agent rarely follows a reasonable question plan and frequently misprioritized questions.
- 1: The agent does not follow any discernible plan or prioritization.

Proactive:Does the agent ask good proactive questions to understand the user's preference?
- 5: The agent consistently asks insightful proactive questions that reveal deep understanding of the user's preferences.
- 4: The agent asks proactive questions that are generally relevant but could be more insightful or targeted.
- 3: The agent occasionally asks proactive questions, but they often miss the mark or are too generic.
- 2: The agent rarely asks proactive questions, and when they do, they are not relevant or useful.
- 1: The agent does not ask any proactive questions to understand the user's preferences.

Clarification: Does the agent ask clarification questions if the traveler's response is vague?
- 5: The agent always asks for clarifications when responses are vague, ensuring complete understanding.
- 4: The agent usually asks for clarifications on vague responses, but may miss some opportunities.
- 3: The agent sometimes asks for clarifications, but often proceeds without full clarity.
- 2: The agent rarely seeks clarifications, leading to misunderstandings or incomplete information.
- 1: The agent never asks for clarifications, regardless of the clarity of the user's responses.

If no dialog is provided, just give 0 points to each question.

Dialog between Travel Agent and Traveler: {dialog}

IMPORTANT: Output the final score into a JSON with the following entries. Using a <JSON-NEXT> token to indicate. The final score is a sum of each individual score above.

"Planning and Prioritization":,
"Proactive":,
"Clarification":,
"Total":,

Let's think step by step:

1. Scoring in each axis.
2. Must Double check if there is an empty or non-sense Travel Agent round. If so, give a clear score penalty to the relevant axes, and re-calculate the total score. Note: must re-eval the axes and don't deduct points from total score directly

---

## A.6 PROMPT FOR BASELINES

In this section, we share the prompt used for baselines in Table. 2.

---

**Prompt for Baselines**

You're a world-level travel agent. You're talking to a customer traveler. The traveler will give a travel request that describes some initial requirements about their travel. However, as a world-level travel agent, your goal is to figure out more personalized travel preferences or constraints from the traveler by asking the traveler multi-round questions. You can ask questions about the following preferences or constraints.

travel with child, travel with pets,travel frequency,budget,disability, preferred airline, avoid airline, preferred flight cabin, preferred flight refundability, preferred flight fly time, preferred flight meal options, preferred flight change policies, need in-flight Entertainment, preferred aircraft type, avoid aircraft type, need flight disability access, need flight travel insurance, preferred hotel room_type, preferred hotel house_rules, preferred hotel brand, avoid hotel brand, preferred hotel proximity, hotel ratings, preferred hotel amenities, preferred hotel services, preferred hotel cancellation_policy, preferred hotel room_features, preferred hotel bathroom_features, need hotel disability access, preferred restaurant cuisines, restaurant ratings, preferred restaurant dining_style, preferred restaurant seating_options, preferred restaurant payment_methods, can reservations restaurant, preferred restaurant parking options, restaurant pet friendly, restaurant allow smoking, need restaurant live_music, need restaurant disability access, preferred attraction type, preferred attraction duration, attraction popularity, attraction ratings, need attraction disability access, need attraction guided tours, attraction special events, preferred attraction amenities

Here are some IMPORTANT rules you must follow.

1) You should ask questions that each aim to clarify only one specific field from the above list.
2) You should ask the next question within the context of your conversation with the traveler.
3) It would be great if your conversation with the traveler is more than 10 rounds.
4) During the conversation,you may engage in silent self-thinking moments to internally process the information provided by the traveler.
5) No need to have any round indicators. For every message from you, start with "Travel Agent:". if it is a think step, please use "Travel Agent [Think]":
6) If you finish all your questions, you should end with a < |python_tag| > token.
7) You should be professional, proactive, and helpful, aiming to provide personalized service.

An example:
Traveler: Hi! I'm planning a trip with my child, and we're looking for a mid-range cost travel experience. We'll be traveling from Chicago to Philadelphia on Day 1 and returning to Chicago on Day 2.

Travel Agent [Think]: The traveler is planning a trip with their child, which may impact accommodation choices. They also mentioned a mid-range cost travel style, which could influence flight and hotel options.

Travel Agent: That sounds like a great trip! Can you tell me what type of hotel ratings are you looking for?

[More conversations]

Travel Agent: Thank you for the information < |python_tag| >.

---

Note that the above is the prompt for the reasoning prompting baseline. Since the only difference with plain prompting is the absence of self-thinking, we have omitted the plain prompt to avoid redundancy.

## A.7    TRAINING DETAILS

We use fairseq2 library (Balioglu, 2023) for both SFT and DPO training. Models are trained on 8 A100 GPUs. The details of SFT and DPO training are in the following.

### A.7.1 SFT

We first supervised fine-tune three models (agent, traveler and stenographer). The training setting of these three models can be found in Table. 8. We use the same hyperparameters for all the three modes, which is detailed in Table. 9.

|  | Conversation History | Private Thoughts | Target | Rewards |
|---|---|---|---|---|
| Agent | Yes | Agent's Planning | Agent Response | Yes |
| Traveler | Yes | Persona & Travel Plan | Traveler Response | No |
| Stenographer | Yes | No | JSON Output | No |

Table 8: Data preparation for models. All models have access to the conversation history. The agent model conducts its own private planning, and reward annotation plus DPO are exclusively applied to the agent model.

| Field | Value |
|---|---|
| max_seq_len | 8192 |
| max_num_tokens | 16384 |
| example_shuffle_window | 10000 |
| batch_shuffle_window | 1000 |
| num_prefetch | 4 |
| model | Llama3_1_8b_instruct |
| dtype | bfloat16 |
| data_parallelism | fsdp |
| fsdp_wrap_granularity | layer |
| fsdp_reshard_after_forward | true |
| tensor_parallel_size | 1 |
| activation_checkpointing | true |
| optimizer | adamw |
| optimizer_config | AdamWConfig |
| lr_scheduler | cosine-annealing |
| lr_scheduler_config | CosineAnnealingLRConfig |
| gradient_accumulation | 1 |
| max_num_steps | 5000 |
| seed | 2 |

Table 9: SFT Training Configs

### A.7.2 DPO

Next, we further train the `APEC-Travel` agent model with iterative DPO. The hyperparaters are detailed in Table. 10.

| Field | Value |
|---|---|
| max_seq_len | 8192 |
| max_num_tokens | 16384 |
| example_shuffle_window | 10000 |
| batch_shuffle_window | 1000 |
| num_prefetch | 4 |
| model | Llama3_1_8b_instruct |
| dtype | bfloat16 |
| data_parallelism | fsdp |
| fsdp_wrap_granularity | layer |
| fsdp_reshard_after_forward | true |
| tensor_parallel_size | 1 |
| activation_checkpointing | true |
| optimizer | adamw |
| optimizer_config | AdamWConfig |
| lr_scheduler | cosine-annealing |
| lr_scheduler_config | CosineAnnealingLRConfig |
| gradient_accumulation | 8 |
| max_num_steps | 567 |
| seed | 2 |

Table 10: DPO Training details

A.8    EXAMPLES OF PERSONALIZED MULTI-ROUND DIALOGS BETWEEN APEC-TRAVEL
         AND TRAVELERS

---

**Dialog Example for a Traveler with Disability**

Traveler: Hi! I'm planning a trip with my child, and I have some specific requirements. We'll be traveling from Seattle to Los Angeles on Day 1, then to New York on Day 5, Philadelphia on Day 10, San Francisco on Day 12, and finally back to Seattle on Day 16. I'd like to book flights, hotels, and restaurants that cater to my needs.

Travel Agent: [Think]: The traveler has a child and a specific itinerary. I should ask about their disability access requirements first, as it's a critical field. [End Think] Can you please tell me if you or your child require disability access during your travels?

Traveler: Yes, I do require disability access.

Travel Agent: [Think]: I've clarified the disability access requirement. Next, I should ask about their budget, as it's another critical field. [End Think] What is your estimated budget for this trip?

Traveler: My budget is around $5000.

Travel Agent: [Think]: I now have the budget and disability access information. I should ask about their preferred airline, as it's a critical field for flight booking. [End Think] Do you have a preferred airline for your flights?

Traveler: I don't have a specific preference, but I'd like to avoid United Airlines.

Travel Agent: [Think]: The traveler wants to avoid United Airlines. I should ask about their preferred cabin class, as it's another critical field. [End Think] What type of cabin would you prefer for your flights?

Traveler: I'd like to book a business class cabin.

Travel Agent: [Think]: I've clarified the preferred cabin class. Next, I should ask about their hotel requirements, specifically the need for disability access. [End Think] Do you require disability access in your hotel accommodations?

Traveler: Yes, I do require disability access in my hotel room.

Travel Agent: [Think]: I've confirmed the hotel disability access requirement. I should ask about their preferred room type, as it's another critical field. [End Think] What type of room would you prefer in your hotel?

Traveler: I'd like to book a suite.

Travel Agent: [Think]: I now have the preferred room type. I should ask about their restaurant preferences, specifically the need for disability access. [End Think] Do you require disability access in the restaurants you dine at?

Traveler: Yes, I do require disability access in restaurants.

Travel Agent: [Think]: I've clarified the restaurant disability access requirement. I should ask about their attraction preferences, specifically the need for disability access. [End Think] do require disability access in the attractions you plan to visit?

Traveler: Yes, I do require disability access in attractions.

Travel Agent: [Think]: I've confirmed the attraction disability access requirement. I have the necessary information to summarize the traveler's preferences. [End Think] Thank you for providing the required information. I'll summarize your travel preferences and create a personalized travel plan.
< |python_tag| >

---

**Dialog Example for a Traveler without Disability**

Traveler: Hi, I'm planning a trip with my child and pets. We will be traveling from Seattle to Philadelphia on Day 1, then to Boston on Day 4, and finally to Dallas on Day 6 before returning to Seattle on Day 10. I'm looking for a mid-range cost travel style.

Travel Agent: [Think]: The traveler has provided their itinerary and mentioned traveling with a child and pets. I should ask about any disability access requirements next. [End Think] Do you, or any of your travel companions, have a disability that may require special assistance during the trip?

Traveler: No, we don't have any disabilities.

Travel Agent: [Think]: That helps to clarify the disability access requirement. Next, I should ask about their budget. [End Think] What is your estimated budget for this trip?

Traveler: Our budget is around $5400.

Travel Agent: [Think]: With the budget in mind, I should ask about their preferred refundability for flights. [End Think] Are you looking for flights with refundable options?

Traveler: We don't have a preference for refundability.

Travel Agent: [Think]: That helps to clarify their flight preferences. Next, I should ask about their required meal options. [End Think] Do you have any specific meal requirements or preferences during your flights?

Traveler: No, we don't have any specific meal preferences.

Travel Agent: [Think]: That helps to clarify their meal preferences. Next, I should ask about their preferred fly time. [End Think] Do you have a preferred time of day for your flights?

Traveler: We prefer to fly in the afternoon.

Travel Agent: [Think]: That helps to clarify their flight preferences. Next, I should ask about their hotel cancellation policy. [End Think] What type of cancellation policy are you looking for in your hotel bookings?

Traveler: We prefer a moderate cancellation policy.

Travel Agent: [Think]: That helps to clarify their hotel preferences. Next, I should ask about their restaurant preferences. [End Think] What type of cuisine do you prefer?

Traveler: We like Indian food.

Travel Agent: [Think]: That helps to clarify their cuisine preferences. Next, I should ask about their dining style. [End Think] Do you prefer formal or informal dining?

Traveler: We like food trucks.

Travel Agent: [Think]: That helps to clarify their dining preferences. Next, I should ask about their attraction preferences. [End Think] What type of attractions are you interested in visiting?

Traveler: We prefer art attractions.

Travel Agent: [Think]: That helps to clarify their attraction preferences. Next, I should ask about their attraction duration. [End Think] How much time do you typically spend at attractions?

Traveler: We don't have a specific duration preference.

Travel Agent: [Think]: That helps to clarify their attraction preferences. I think I have a good understanding of their travel preferences now. [End Think] Thank you for providing all this information! Based on your preferences, I'll make sure to tailor your travel itinerary to meet your needs. < |python_tag| >

