# OpenReview forum: "Towards Full Delegation: Designing Ideal Agentic Behaviors for Travel Planning"
_ICLR.cc/2025/Conference — Submitted to ICLR 2025_

### Official Review · Reviewer_UTGM · 2024-10-28

**Soundness:** 3
**Presentation:** 4
**Contribution:** 3
**Rating:** 6
**Confidence:** 4

**Summary:**

The paper proposes a framework, APEC, defining criteria for good agentic behavior termed the "Agent Constitution." APEC-Travel is introduced as a running scenario, demonstrating the framework's applicability to travel planning, with agents trained using DPO to align with APEC principles. The work utilizes rule-based metrics and agentic scores for training, and synthetic data generation creates 54 personas informed by a human survey for personalization in travel planning.

**Strengths:**

1. **Novelty** : The concept of an "Agent Constitution" is compelling and could serve as a foundation for designing ethical and efficient agent behaviors across various applications.
2. **Empirical Validation (Quality)**: APEC-Travel demonstrates the framework’s applicability and effectiveness in generating coherent travel plans, which is evaluated through agentic scores and rule-based metrics.
3. **Personalization Significance**: The introduction of 54 empirically supported personas is a thoughtful addition to enhancing travel planning experiences.
4. **Clarity**: I find few minor details missing with gap in definition and implementation of the metrics. Although the text is well written to explain the major claims made.

**Weaknesses:**

The paper provides an interesting perspective on establishing agentic principles and has shown the potential for real-world impact, especially in travel planning. Addressing the points mentioned could enhance its clarity, rigor, and applicability to broader domains.

1. Validation of Synthetic Personas
    * Clarification Request: The paper lacks clarity on whether the simulated personas were validated beyond their construction from survey results. Given the random entry assignment for synthetic traveler generation, it would be valuable to know if validation methods (e.g., similarity scoring to real user behavior) were employed or considered.
2. Switch from Large to Smaller Model During “Critical Persona Entry Selection”
    * Justification Needed: During Seed dialogue generation and agentic scoring, a larger model (Llama3.1-405B-Instruct) was used, likely for accuracy. However, during the "Critical Persona Entry Selection," a smaller version (Llama3.1-8B-Instruct) was utilized. Is there a performance or computational rationale for this model switch, and how does it affect outcome consistency?
3. Efficiency Definition and Mapping to Experimental Context
    * Clarification Request: The definition of efficiency as "accuracy gain per round" does not seem to align with the context described before the experiments, where it mentions "checking whether the agent has obtained critical information with limited dialog exchanges." Was efficiency calculated by invoking the Stenographer after each round to validate information retrieval or solely by accuracy differential per round? Additional clarification on this could help contextualize the metric’s relevance.
4. Exclusion of Efficiency in Reward Calculation
    * Inquiry: Why was efficiency, a key metric, excluded from the reward calculation for DPO training, despite its apparent relevance in performance benchmarking? Is it because its derived from Accuracy?
5. Sentence Construction in Section 4.3
    * Suggestion: Minor improvements in sentence structure are needed just before Section 4.3. For example, "suggesting the efficacy of our training recipe in consistently train agents that align with the principles..." could be improved.
6. Alpha Value in Reward Calculation
    * Clarification Request: The value of alpha in the reward calculation appears significant, yet it's unclear what alpha was set to for the primary results shown in Table 2. This information would aid reproducibility and could be noted in the table or text if it was overlooked.
7. Alpha Setting in Iterative Training Paradigm
    * Inquiry: In the ablation study, "Iterative Training Paradigm" shows only accuracy improvements between fixed and recursive paradigms. Was this improvement due to a high alpha value in the reward calculation? Further details would clarify the choice and impact of alpha in iterative training.
8. Evaluation Across Domains and Real-World Datasets
    * Suggestion: While APEC demonstrates effectiveness in APEC-Travel, additional evaluation on other agentic domains (e.g., web exploration, embodied agent tasks) would strengthen claims of generality. Additionally, employing established travel planning datasets (e.g., TravelPlanner[Xie et al. 2024b]) could enhance validation beyond synthetic data. I assume this as a limitation of this work.

**Questions:**

I have listed the questions, concerns and points required for clairty above and I am looking forward to the rebuttal during the disscussion phase.

---

> ### Author Response · Authors · 2024-11-20
> **Response to Reviewer UTGM (1/2)**
>
> We sincerely appreciate your positive review and the valuable feedback you've provided. Your insights are instrumental in helping us refine our work. In the following, we provide explanations and clarifications to your concerns one-by-one. We really appreciate your suggestions and hope our response could address your concerns.
>
>
> > **W1: Validation of Synthetic Personas**
>
> **A1:** We would like to clarify that one major consideration in persona simulation is to ensure the diversity of simulated travelers. We achieve this goal by the random assignment of traveler entries. That being said, we indeed have two efforts to ensure the validity of the simulated traveler. 1) During traveler persona simulation, we utilize a rule-based post-process to filter out the unreal travelers. Specifically, we design rules based on common-sense of travelers. For example, if a traveler has a disability, then a ‘Yes’ must be assigned to the “need disability access” entry in flight need.  2) The critical entries are derived based on a user survey of 1385 travelers, which further aligned the persona with real travelers. In addition, In response to your feedback, we have conducted additional experiments to further demonstrate the validation of simulated personas. Specifically, from the 54 travel persona entries, we select 6 important ones, including age, education, disability, budget, service quality, and traveling with pets to analyze whether our APEC-Travel effectively accommodates personalized personas. We present these new results in Figure 9 in the updated draft pdf. Our new results demonstrate that our APEC-Travel agent is robust to various traveler types and consistently outperforms the baselines.
>
> > **W2: Switch from Large to Smaller Model During “Critical Persona Entry Selection**
>
> **A2:** We understand your concern about using a smaller model for critical persona entry selection. We would like to clarify that we mainly use a larger model (Llama3.1-405B-Instruct) for seed dialog generation and agentic scoring, because these two are more difficult tasks that involve longer and more complex understanding and generation. Our early experiments found it hard to achieve satisfied results if using a small model (e.g., Llama3.1-8B-Instruct) on these two tasks. However, critical persona entry selection is a relatively easier task and we find a smaller model is already good enough and can generate a reasonable rank of persona entries. For example, we observe that both Llama3.1-405B-Instruct and Llama3.1-8B-Instruct agree that for a traveler with disability, the “need disability access” in flight, hotel, restaurant and attraction would be in the critical priorities, similarly, a for a traveler cares total price, the ‘budget’ entry will be ranked very high (i.e., one of critical entries) by both the 405B and 8B models. Therefore, we keep the small model for critical persona entry selection in our experiments.
>
> > **W3: Efficiency Definition and Mapping to Experimental Context**
>
> **A3:** Thanks for the detailed question. We appreciate the opportunity to clarify this aspect of our work. In our experiments, the "efficiency" metric is defined as the accuracy gain per round, which reflects how quickly and accurately the agent can gather necessary information to achieve its goals, as outlined in the APEC Agent Constitution. This metric is crucial for evaluating the agent's ability to obtain critical information with minimal dialog exchanges.
> To calculate efficiency, we assess the agent's performance after each dialog round. Specifically, we measure the accuracy of information collection at each step and determine the gain in accuracy compared to previous rounds. This approach allows us to quantify how effectively the agent is progressing towards its objectives with each round.
> We report this efficiency metric for both overall and critical traveler persona entries. A higher efficiency value indicates that the agent is not only collecting information accurately but also doing so in fewer dialog exchanges, thereby demonstrating its ability to prioritize and focus on decision-relevant information.

---

> ### Author Response · Authors · 2024-11-20
> **Response to Reviewer UTGM (2/2)**
>
> > **W4: Exclusion of Efficiency in Reward Calculation**
>
> **A4:** Thanks for the insightful question. Our reward contains Accuracy and Agentic Scores (Plan & Priority, Proactive, Clarification). We believe efficiency is already implicitly captured by these metrics. Specifically: 1) Plan & Priority evaluates the agent's ability to prioritize critical entries and effectively plan the sequence of questions. By ensuring that agents focus on the most critical entries, this metric inherently promotes efficient information collection. 2) Proactivity and Clarification assess the agent's ability to take initiative and seek clarification when needed, further contributing to efficient dialog management. Our experimental results in Table 2 also demonstrate that aligning agents with rewards based on Accuracy and Agentic Scores consistently improves the efficiency metric. This indicates that the existing reward components effectively drive the agent towards efficient performance without explicitly including efficiency as a separate metric.
>
> > **W5: Sentence Construction in Section 4.3**
>
> **A5:** We appreciate your attention to detail and agree that clarity and readability are crucial for effectively communicating our experiment findings. We have revised the sentence to enhance its clarity and flow. The updated sentence is:
> "The results demonstrate that the iteratively trained APEC-Travel-DPO, especially in iteration 2, significantly enhances the agentic scores across the entire test set. This improvement underscores the effectiveness of our training recipe in consistently aligning agents with the principles in the Agent Constitution."
>
> > **W6: Alpha Value in Reward Calculation**
>
> **A6:** Thanks for the detailed suggestion. In our experiments, the alpha value was set to $\alpha$=0.1 for the primary results presented in Table 2. We have added this in our revised paper to aid in reproducibility and provide clarity for future readers.
>
> > **W7: Alpha Setting in Iterative Training Paradigm**
>
> **A7:** In the ablation study, we maintain a constant alpha value of $\alpha$=0.1 to isolate the effects of the training paradigm itself. By keeping alpha fixed, we can ensure that the improvements in accuracy between the fixed and recursive paradigms were not influenced by changes in the reward calculation's weighting. Therefore, we can conclude that the improvements are attributed to the differences in training approaches rather than the alpha value. This design allows us to focus on evaluating the impact of the iterative training process on agent performance.
>
>
>
> > **W8: Evaluation Across Domains and Real-World Datasets**
>
> **A8:** Thank you for your insightful comments.  We agree that evaluating our approach in additional agentic domains could further demonstrate the effectiveness of the APEC Agent Constitution. However, we would like to highlight that the principles outlined in the APEC Agent Constitution, along with our training recipe and reward construction, are designed to be general and not specific to travel agents. These principles are intended to be naturally and directly applicable to a wide range of agent environments.
> They are general principles that can be naturally and directly applicable to other agent environments.
> We acknowledge the value of established datasets like TravelPlanner [Xie et al. 2024b] in evaluating planning capabilities. However, we’d like to clarify that this benchmark assumes all constraints are explicitly provided in one shot, which may not reflect real-world scenarios where constraints typically emerge through multi-round dialogs between agents and users. As discussed in the related work section, our approach is also inspired by this work and we realize that human-like agentic behaviors beyond just measuring final outcome in TravelPlanner is a further step towards full delegation agents.
>
> Thank you again for your valuable feedback, which helps us refine our research and consider new directions for future work. We hope that our clarifications and additional results address your concerns and demonstrate the value of our contributions.
>
> Reference:
>
> [1]  Xie et al 2024, TravelPlanner: A Benchmark for Real-World Planning with Language Agents

---

> > ### Comment · Reviewer_UTGM · 2024-11-26
> > **Incorporate limitations in this work**
> >
> > Dear Authors, Thankyou for handling queries. I would request to note following points in a limitation section, if paper gets accepted
> > * Concern mentioned in W2, as “we find a smaller model is already good enough” is quite subjective in nature and there should be consistency in the base model for data generation.
> > * Regarding W4: “.....efficient performance without explicitly including efficiency as a separate metric.”, but analysis remains to include it as a different parameter during reward calculation. So there is scope to include more parameters to determine reward aligned to the task in hand, which can help better identify the appropriate reward score.
> > * General applicabilityW8: TravelPlanning dataset is a good start, but its a multi-day single-POI(point of interest) travel planning, there are more complex and wider datasets such as Single day Multi-POI TravelPlanning[1] and Multi-day multi-POI TravelPlnanning[2]. Over and above these all are offline problem solving tasks while embodied agents or other environments may offer online problem solving where things are more dynamically changing for action exploration while the agent traverses the environment. Therefore a word of caution seems a necessity.
> > * Exploration of other suitable metrics for tasks differing in wider perspective than TravelPlanning to have few other.
> >
> > I keep my score unchanged for now. Thankyou for additional information.
> >
> > References:
> > [1]: Tang, Yihong, et al. "Synergizing Spatial Optimization with Large Language Models for Open-Domain Urban Itinerary Planning." arXiv preprint arXiv:2402.07204 (2024).
> > [2]: Chen, Aili, et al. "TravelAgent: An AI Assistant for Personalized Travel Planning." arXiv preprint arXiv:2409.08069 (2024).

---

> > > ### Author Response · Authors · 2024-11-28
> > >
> > > We sincerely thank you for your reply and further detailed feedback, especially the additional related papers, We will for sure include them in our next version of draft.
> > >
> > >
> > > Authors

---

### Official Review · Reviewer_cZwH · 2024-10-29

**Soundness:** 3
**Presentation:** 2
**Contribution:** 2
**Rating:** 3
**Confidence:** 3

**Summary:**

The paper proposes a "constitution" for evaluating the behavior of LLM agents beyond accuracy. In particular, it proposes four criteria: (1) accuracy, (2) proactivity, (3) efficiency, and (4) credibility. In a travel planning setting, the paper then uses prompting-based methods to score agent trajectories on these four categories, resulting in an "accuracy score" and an "agent score" which are combined to form a reward measure. The paper then shows a proof-of-concept that using SFT on data from a stronger model, as well as DPO, scores on this reward can be increased substantially.

**Strengths:**

1. I think the general idea of evaluating agents beyond accuracy is reasonable and well-motivated. There are many aspects of what makes a good user experience (e.g., did the agent provide compelling evidence for its decisions?) that aren't necessarily reflected in task success metrics

2. The experiments are thorough, easy to follow, and include several ablations. Additionally, the approach to persona construction seems reasonable and relatively well-aligned to the preferences of travelers in the real world

**Weaknesses:**

1. The proposed "constitution" lacks a clear motivation for why it chooses the four principles (accuracy, proactivity, efficiency, credibility) that it does. In particular, it's unclear to me why you need all four of these principles: can an agent be accurate and efficient without also being proactive and credible? Additionally, the main experiments in the paper use a different set of principles (plan & priority, proactive, clarification), calling into question the utility of the original set of four

2. The method seems to be primarily focused on distilling the behavior of a stronger model (Llama3.1-405B-Instruct) into a weaker model (Llama3.1-8B-Instruct). As a result, I don't find the performance improvements in this paper to be very surprising, and it's not clear to me that they teach us anything new. It would be nice to have clearer takeaways from the experimental work

**Questions:**

1. As far as I can tell, the proposed task is less about actual travel planning (i.e., what is the quality of the proposed itinerary?) and more about preference elicitation and inference. In the real world analogue of this setting, it seems evident that many preferences can be elicited that aren't decision-relevant. How does your approach to evaluation account for this issue?

---

> ### Author Response · Authors · 2024-11-20
> **Response to Reviewer cZwH  (1/2)**
>
> We sincerely appreciate reviewer cZwH for the thoughtful review and for highlighting areas where our work can be improved. We provide explanations to your concerns point-by-point in the following. We hope that these new results and explanations adequately address your concerns, and we welcome further discussion.
>
> > **W1 (a): Motivation of four principles in APEC Agent Constitution. In particular, can an agent be accurate and efficient without also being proactive and credible?**
>
> **A1 (a):** We sincerely thank reviewer cZwH for this detailed question and for recognizing the importance of evaluating agents beyond mere accuracy in the “Strengths” part.  Therefore, it would be essential and natural to incorporate more aspects of agentic behaviors into the Agent Constitution.
>
> Specifically, we’d also like to further clarify our motivation and vision of this paper is to build *full-delegation* agents: they can take over humans’ routine decision-making processes and find solutions that fit people’s personalized needs. We argue that enabling humans to delegate is a key property of LLM agents. To achieve this goal,  LLM agents should be evaluated and optimized not only based on the final outcome (e.g., accuracy on benchmarks), but also the procedure of how agents achieve the goal (proactivity, efficiency, credibility). Having such agentic behaviors is also a crucial step towards building human-like agents. Taking our travel agent as an example, beyond the final accuracy of inferring travelers’ preferences, the travel agent must prioritize critical questions according to the traveler’s personalized request (efficient, credibility) and be proactive to move forward with new topics to gain more travel information (proactive).
>
> > **W1 (b): Different set of principles (plan & priority, proactive, clarification)?**
>
> **A1 (b):** We’d like to clarify that APEC (Accuracy, Proactivity, Efficiency, Credibility) is our four principles of full-delegation agents, while (plan & priority, proactive, clarification) are concrete and quantitative metrics proposed for evaluating the APEC principles. For example, "Efficiency'’ in APEC is an abstract concept; to quantify it for both optimization and evaluation purposes, we use "plan & priority’’ and "clarification’’ as specific metrics assessed by models. The details of how these metrics are used to evaluate APEC are provided in Table 1 of the paper (also included below).
>
> ||  Accuracy | Proactiveness | Efficiency | Credibility |
> |:-----------:|:----------:|:------------------------:|:---------:|:---------:|
> |Symbolic (JSON)| ✅      |                   | ✅ | |
> |Plan & Priority|       |                    |   ✅| ✅ |
> |Proactive|      | ✅                   |   | |
> |Clarification|     |                    | ✅ |  |
>
>
>
> We also would like to highlight that in order to refine and validate these metrics, we conducted a human annotation experiment, in which 58 human annotators from a reputable third-party data annotation company are involved to score 200 sample dialogs between agents and travelers (detailed experiment setup is presented in section 3.1 and appendix. A.4). The results from human annotation demonstrate that these metrics are independent of each other and effectively reflect the APEC principles.
>
>
> > **W2: Distilling the behavior of a stronger model (Llama3.1-405B-Instruct) into a weaker model and Clearer takeaways from the experimental work**
>
> **A2:** We understand your concern regarding knowledge/behavior distillation between models. However, we’d like to clarify that our work goes beyond simple knowledge distillation by introducing a comprehensive training recipe for developing full delegation agents. One reason for using the Llama3.1-405B model is to ensure that our training recipe is applicable across various domains, even when large-scale human-annotated data is not available. In this context, the Llama3.1-405B model serves as a synthetic data source, enabling effective agent training without the need for extensive human annotations.
>
> Furthermore, designing rewards to align LLM agents with both accurate solutions and agentic behaviors is a complex challenge that cannot be straightforwardly addressed by distillation from Llama3.1-405B. Our approach provides a structured method to achieve this alignment.

---

> ### Author Response · Authors · 2024-11-20
> **Response to Reviewer cZwH  (2/2)**
>
> **A2 (Cont'd):**
>
> **Key Takeaways from Experiments**:
>
> *Training Recipe Effectiveness*: Our experiments demonstrate that our proposed training recipe successfully aligns agents with the agentic behaviors outlined in the constitution, moving towards the goal of full-delegation agents. This alignment is a significant step forward in developing autonomous agents capable of complex decision-making.
>
> *Reward Model Design*: The integration of both symbolic metrics (JSON accuracy) and neural metrics (LLM-as-a-judge for agentic scores) in our reward model is pivotal. This dual approach ensures alignment with the agentic behaviors outlined in the constitution, providing a balanced evaluation of both accuracy and agentic behavior.
>
> *Efficiency in APEC-Travel Agent*: Our breakdown analysis of the APEC-travel agent highlights the essential role of DPO with our reward model in the training recipe. In addition, our ablation study reveals that during DPO, using the model from latest iteration as the reference model can lead to an overemphasis on certain sub-goals, such as accuracy (0.279->0.320 ⬆️), at the expense of others, like agentic scores (12.95->12.10 ⬇️) . Instead, by using the SFT model as the reference model and generating dialogs with the latest model in each DPO iteration, we mitigate this limitation, ensuring a more balanced alignment with multiple goals.
>
> > **Q1: How to evaluate decision-relevant preferences?.**
>
> We also acknowledge and recognize the importance of distinguishing between decision-relevant and non-relevant preferences in the context of travel planning. In APEC-Travel, we specifically addressed this challenge by ensuring that our agents focus on identifying and prioritizing critical user preferences.
>
> Specifically, each traveler has a unique set of priorities related to their travel preferences. For instance, a traveler with a disability may prioritize accessible services above all else. To ensure realistic and critical persona entries, we factor in basic user characteristics such as age, job, education, marital status, disability, and travel style, along with empirical traveler categories. This approach is grounded in an online survey of 1,385 travelers. To create critical entries for a given user, we prompt the LLM to rank the importance of the full set of persona entries and select the top 20% as critical entries (details in Section 3.2 and Appendix A.2). Our agent seeks to identify these critical entries through multi-round dialog.
>
>
> During evaluation, instead of just the overall preference inference accuracy, we also report the accuracy on critical entries and assess how efficiently agents identify these entries (as shown in Table 2). This dual focus ensures that our evaluation captures both the relevance and efficiency of preference elicitation.
>
> In addition, our approach consolidates all inferred travel preferences into a symbolic format (JSON). We want to highlight that evaluating these inferred travel preferences is a crucial step as in existing work [1-2]. Once we have accurately inferred the JSON, it can be directly input into a symbolic solver, as demonstrated in [1-2], to generate the travel itinerary. Therefore, evaluating the inferred JSON would already allow us to assess how effectively our approach captures decision-relevant preferences. We also acknowledge that calling a symbolic solver to convert our inferred JSON into a travel itinerary is a straightforward and immediate next step (though it needs real-time travel data APIs, e.g., flight API, hotel API, etc), which we leave for future work. We’ve added this discussion in the revised pdf.
>
>
>
>
> We hope that our clarifications and additional insights address your concerns and demonstrate the value of our contributions to the community. We are eager to engage in further discussion and are open to making additional revisions to enhance the clarity and impact of our work.
>
>
> Reference:
>
> [1]  Da Ju, Song Jiang, Andrew Cohen, Aaron Foss, Sasha Mitts, Arman Zharmagambetov, Brandon Amos, Xian Li, Justine T Kao, Maryam Fazel-Zarandi, and Yuandong Tian. *To the Globe (TTG): Towards Language-Driven Guaranteed Travel Planning*. EMNLP 2024 Demo Track .
>
> [2] Yilun Hao, Yongchao Chen, Yang Zhang, Chuchu Fan. "Large Language Models Can Plan Your Travels Rigorously with Formal Verification Tools." arXiv preprint arXiv:2404.11891 (2024).

---

> ### Comment · Reviewer_cZwH · 2024-11-25
>
> > We’d like to clarify that APEC (Accuracy, Proactivity, Efficiency, Credibility) is our four principles of full-delegation agents, while (plan & priority, proactive, clarification) are concrete and quantitative metrics proposed for evaluating the APEC principles
>
> I am still unconvinced that the set of APEC principles are useful, and don't see the need to have one set of "abstract" principles and another set of "concrete" principles. In fact, I would argue that this statement suggests that the more concrete properties are more useful than the APEC properties. Additionally, I am still concerned that the APEC properties aren't sufficiently defined, and are redundant with one another.
>
> > We understand your concern regarding knowledge/behavior distillation between models. However, we’d like to clarify that our work goes beyond simple knowledge distillation by introducing a comprehensive training recipe for developing full delegation agents
>
> I also think it's not clear to me what the exact novelty of this training recipe is, e.g., compared to methods like Constitutional AI.
> ***
> I appreciate the detailed responses, but I'll keep my score the same for now.

---

> > ### Author Response · Authors · 2024-11-28
> > **Follow-up Response to Reviewer cZwH (1/2)**
> >
> > We sincerely appreciate reviewer cZwH for providing additional follow-up comments. Below, we address each of your concerns with detailed explanations and clarifications. We value your suggestions and hope our responses effectively address your concerns.
> >
> > > **1. Have one set of "abstract" principles and another set of "concrete" principles. In fact, I would argue that this statement suggests that the more concrete properties are more useful than the APEC properties**
> >
> > **A1:** We’d clarify that our intention is not to propose separate "abstract" and "concrete" sets of principles. Instead, we utilize a set of metrics—Symbolic (JSON) Accuracy, Plan & Priority, Proactivity, and Clarification—as measurements to evaluate how well an agent adheres to the APEC agent constitution. These metrics are designed to operationalize the abstract principles within specific contexts.
> >
> > Taking “efficiency” in APEC as an example, to measure how efficient an agent is, we need metrics. In the travel agent context, one key perspective of efficiency is whether the travel agent can prioritize critical personalized entries and plan the sequence of questions. Therefore, we include Plan & Priority score as one measurement to evaluate an agent’s efficiency ability. (Note we also use metrics from Symbolic (JSON) Accuracy, and the Clarification score as additional measurements for a comprehensive evaluation of efficiency).
> >
> > In fact, for many machine learning tasks, it would be natural to use concrete metrics to evaluate a principle. For example, when evaluating accuracy, researchers use “F1 score”, “AUC”,”pass@k” and etc according to specific contexts. Therefore, it is not a matter of comparing the usefulness of APEC principles versus these concrete metrics; rather, the metrics serve as instruments to measure the principles' application.
> >
> >
> > > **2.** *Here we combine your two comments of* **“I am still unconvinced that the set of APEC principles are useful” and “ APEC properties aren't sufficiently defined, and are redundant with one another”** *since they are both about APEC principles*
> >
> >
> > **A2:** We’d  like to emphasize our overarching goal: to develop full-delegation agents. This requires evaluating agents on dimensions beyond mere accuracy, as a single accuracy metric cannot capture the multifaceted requirements necessary for real-world utility. Adhering to the APEC principles is a crucial step toward achieving this goal.
> >
> > *On the Sufficiency of APEC Principles:* We agree that various aspects can be proposed depending on specific contexts. However, the scope of our paper is not to provide an exhaustive set of rules for all different real-world scenarios. Instead, we aim to identify the most common principles essential for full-delegation agents. Our APEC framework—comprising Accuracy, Proactivity, Efficiency, and Credibility—addresses key common dimensions:
> > - Accuracy: Ensures the quality of the final solution.
> > - Proactivity: Involves actively gathering information to solve tasks.
> > - Efficiency: Focuses on achieving goals with minimal interactions.
> > - Credibility: Relates to the reliability and positive outcomes of task execution.
> >
> > These principles focus on process, outcome, and reliability of the routine of agents execute tasks. While we demonstrate APEC using a travel agent example, these principles are broadly applicable. For instance, in coding agents, efficiency might involve planning code structure effectively and minimizing external API calls.
> >
> > *Addressing Redundancy Concerns:* We’d direct the reviewer to Section 3.1 of our paper, where we detail our human annotation process conducted via a third-party vendor. This process was instrumental in reducing redundancy and ensuring the independence of our principles. Initially, we proposed a broader set of measurements, which were refined through human annotation to focus on the most distinct and essential aspects. Our experiments, as shown in Figure 8, validate the significant reduction in redundancy achieved through this approach.

---

> > > ### Author Response · Authors · 2024-11-28
> > > **Follow-up Response to Reviewer cZwH (2/2)**
> > >
> > > > **3. I also think it's not clear to me what the exact novelty of this training recipe is, e.g., compared to methods like Constitutional AI**
> > >
> > > **A3:** Both our training recipe and Constitutional AI involve SFT and preference optimization stages. Nowadays this framework has indeed become a standard in developing language model-based chat assistants and agents. One unique part in our pipeline is the integration of reward metrics. Specifically, we combine symbolic metrics (such as JSON accuracy) with neural metrics (using LLM-as-a-judge for agentic scores). This integration allows us to create preference pairs that blend objectively verifiable scores with those derived from Constitutional AI. Our efforts in the design of the LLM-as-a-judge rubrics ensures that these two types of scores address different aspects of agent performance. This comprehensive evaluation effectively aligns with the agentic behaviors outlined in our constitution
> > >
> > > Additionally, we address a limitation in the standard iterative DPO procedure, which typically uses the latest model as the reference model. We found that this approach can lead to an overemphasis on certain sub-goals in building agents with multifaceted goals, such as increasing accuracy (0.279->0.320 ) at the expense of others, like agentic scores (12.95->12.10 ) . To address this, we fix the SFT model as the reference model, iteratively tuning it with dialogs generated by the latest DPO model in each iteration. Our experiments, as shown in Table 4, demonstrate the effectiveness of our this design in building agents with multifaceted goals.
> > >
> > >
> > >
> > > We sincerely appreciate your valuable feedback. We hope that our clarifications could effectively address your concerns. We would be grateful for any additional feedback and welcome further discussion.

---

> > > > ### Author Response · Authors · 2024-12-02
> > > > **End of author-reviewer discussion**
> > > >
> > > > Dear Reviewer cZwH,
> > > >
> > > > We sincerely appreciate the time and effort you've devoted to reviewing our work. We understand that your schedule may be quite busy. As the authors-reviewer discussion phase draws to a close, we kindly request your attention to our follow-up response. Our aim is to gain insights into whether our response effectively addresses your concerns and to ascertain if there are any additional questions.
> > > >
> > > > We look forward to the opportunity for further discussion with you. Thank you for your thoughtful consideration.
> > > >
> > > > Best regards,
> > > > Authors

---

### Official Review · Reviewer_AaCo · 2024-11-09

**Soundness:** 3
**Presentation:** 4
**Contribution:** 3
**Rating:** 6
**Confidence:** 4

**Summary:**

This paper introduces  full delegation, where individuals routinely entrust their routine decision-making to agents capable of making high-quality decisions in real-world scenarios tailored to personal needs. To achieve this, the paper has designed the APEC Agent Constitution, where APEC stands for Accuracy, Proactivity, Efficiency, and Credibility. The paper assesses both the process and the final outcome, and develops APEC-Travel to actively gather personalized travel preferences from users via multi-round dialogue. The experimental results on synthetic seed dialogues demonstrate impressive performance when compared to existing baselines.

**Strengths:**

The paper is well-written and organized. It carefully selects 200 examples from the seed dialogues generated by the llama3.1-405b-instruct model, with human annotators evaluating them across five metrics: planning, prioritization, proactiveness, clarification, and helpfulness. The concrete measurement of each axis convincingly contributes to the construction of the final reward construction.

The experimental results show that APEC-Travel-DPO enhances agency scores and efficiency, leading to more concise dialogues with fewer interactive turns.

The ablation study is comprehensive, covering various aspects and different hyper-parameter settings. Additionally, error analysis on 20 examples provides valuable insights into the factors contributing to low-quality dialogues.

**Weaknesses:**

While guidelines are provided for human annotators on how to assess the dialogues, there is a lack of detailed information regarding the number of annotators involved, the process of ensuring consistency through cross-validation, and other measures taken to ensure the reliability of the final scores.

In the evaluation, although the test set comprises 1,000 examples with unique personas, there is a concern that all the seed dialogues (10k), reward construction examples (1k), and test examples (1k) are generated using the same llama3.1-405b-instruct model. This raises the possibility that the training and evaluation dialogues may still share similar distributions, except for persona variations.

**Questions:**

Thanks for providing the basic training configurations for SFT and DPO in Tables 9 and 10. I noticed that increasing the number of iterations in DPO training could deteriorate performance. Could you please share the learning rates used for SFT and each stage of DPO? Additionally, do you have further insights beyond what is discussed in the "Iterative Training Paradigm" section?

---

> ### Author Response · Authors · 2024-11-20
> **Response to Reviewer AaCo**
>
> We sincerely appreciate your positive comments and really thank you for the in-depth and insightful suggestions and questions! We provide a response one by one as follows. We hope these new results and explanation could address your concerns and we welcome further discussion.
>
>
>
>
> > **W1: Detailed information regarding the human annotation.**
>
> **A1:** We hired a reputable third-party data annotation company to conduct the seed dialogue annotations. This company is well-regarded in the AI industry, and we can disclose their name upon obtaining legal approval. Specifically, 58 annotators from the United States participated in our annotation process of 200 seed dialogs, with most annotators scoring between 10 to 15 dialogs. To ensure consistency and minimize variance, each dialog was annotated by 3 different annotators. Additionally, to further ensure reliability, annotators were asked to provide their rationale for each score assignment. By reviewing the rationale behind each score, we can confirm that the scores are both reasonable and reliable. Below is an example of an annotator who assigned a score of 4 out of 5 to a dialog on the clarification axis:
>
> *Clarification score: 4. Rationale: The Travel Agent did a great job at asking many clarification questions, making sure that no stone was left unturned. It noted that the Traveler needed disability access and started to ask in-depth on what exactly would be needed from every destination and activity to better accommodate the Traveler, making their trip a lot smoother when planned by the Travel Agent.*
>
> We've provided this information in our revised paper to ensure clarity of our human annotation process.
>
>
>
>
>
>
> > **W2: Distribution difference between training and evaluation dialogs.**
>
> **A2:** We’d like to clarify that the 10k training (seed) dialogs were generated by the Llama3.1-405B-Instruct model. However, the 10K reward construction dialog and 1k test dialogs were generated by our trained APEC-Travel models in the corresponding steps (SFT or DPO), instead of Llama3.1-405B-Instruct. Therefore, the distributions of training and evaluation dialogs are not the same. This difference can also be reflected from some statistics of the dialogs (number of rounds): training /  evaluation (DPO iter 1) /  evaluation (DPO iter 2) = 10.84/11.19/9.77.
>
> > **Q1: Learning rates used for SFT and each stage of DPO.**
>
> **A1:** Thanks for the detailed question. We use the default learning rate from Fairseq2 codebase (https://github.com/facebookresearch/fairseq2) for both SFT and all DPO stages. Specifically, the lr is 5.5e-06 with a cosine-annealing learning rate scheduler (also implemented by Fairseq2).
>
> > **Q2: Further insights about Iterative Training Paradigm.**
>
> **A2:** We presented our major findings about Iterative Training Paradigm, including impacts of reward construction, choice of reference model and performance of each stage in sections 4.2 and 4.3.
> We add a new experiment to understand if the DPO trained agent APEC-Travel can effectively accommodate personalized personas. Specifically, from the 54 travel persona entries, we select 6 important ones , including age, education, disability, budget, service quality, and traveling with pets. We present the detailed results in Figure. 9 in the revised paper. Our results indicate that DPO trained agent APEC-Travel is robust to various traveler types and consistently outperforms the baselines.
>
>
>
> Thank you once again for your constructive feedback. We hope this explanation could further address your comments.

---

> > ### Comment · Reviewer_AaCo · 2024-11-26
> > **Official Comment by Reviewer**
> >
> > Thanks for your detailed reply. After reviewing the responses, I maintain my current positive score.

---

### Meta-Review · Area_Chair_uT7q · 2024-12-21

**Metareview:**

The paper argues that agents should be evaluated beyond just task success rates and accuracy scores,
and proposes an "Agent Constitution" that aims to assess Accuracy, Proactivity, Efficiency and Credibility.
The reviewers agreed that the problem of aligning agents is well motivated and timely,
though there was significant disagreement about whether the paper's contributions are above the bar for publication at ICLR.
The reviewers appreciated the novelty of the proposal, and the demonstrated results on a travel planning task,
but were unconvinced by the transferability of the findings, as well as the necessity & sufficiency of the proposed constitution.

1. The paper states APEC as necessary dimensions for agent evaluation, as an axiom.
There needs to be more justification. There have been other principles for LLM-based chatbots (e.g. helpfulness and harmlessness, to name two).
How do the APEC principles relate to those studied for chatbots? What are the specific principles that are motivated by agentic behavior? Why can we not add those principles to the set of principles elucidated for chatbots?
Are APEC a sufficient set of dimensions for evaluating agents? Are there other principles that have not been captured, for instance,
all side-effects caused by the agent are made apparent to the user (e.g. some form of scrutability).

2. Reviewers expected experiments on TravelPlanner datasets and more complex agent tasks, to build confidence that the proposed set of APEC principles
indeed capture desirable behavior across different agent benchmarks.

**Additional Comments On Reviewer Discussion:**

The authors highlighted an interesting finding from their experiments that suggests an improvement to the DPO training recipe (i.e. do not use the most recent model for fine-tuning, when aiming for multi-objective alignment). They also described experiments to validate the persona synthesis for their travel agent.
However, the two primary weaknesses identified by reviewers above remain unaddressed. Addressing them will substantially strengthen the paper.

---

### Decision · Program_Chairs · 2025-01-22

Reject